Journal of Data-centric Machine Learning Research (2024)    Submitted 11/23; Revised 1/24; Published 3/24

# Detecting Errors in a Numerical Response via any Regression Model

**Hang Zhou**                                                                    HGZHOU@UCDAVIS.EDU
*Department of Statistics*
*University of California*
*Davis, CA 95618, USA*

**Jonas Mueller**                                                               JONAS@CLEANLAB.AI
*Cleanlab*
*San Francisco, CA 94110, USA*

**Mayank Kumar**                                                    MAYANK.KUMAR@CLEANLAB.AI
*Cleanlab*
*San Francisco, CA 94110, USA*

**Jane-Ling Wang**                                                        JANELWANG@UCDAVIS.EDU
*Department of Statistics*
*University of California*
*Davis, CA 95618, USA*

**Jing Lei**                                                            JINGLEI@ANDREW.CMU.EDU
*Department of Statistics and Data Science*
*Carnegie Mellon University*
*Pittsburgh, PA 15213, USA*

**Reviewed on OpenReview:** *https://openreview.net/forum?id=CIQ5iemeTw*

**Editor:** Raul Castro Fernandez

## Abstract

Noise plagues many numerical datasets, where the recorded values in the data may fail to match the true underlying values due to reasons including: erroneous sensors, data entry/processing mistakes, or imperfect human estimates. We consider general regression settings with covariates and a potentially corrupted response whose observed values may contain errors. By accounting for various uncertainties, we introduced veracity scores that distinguish between genuine errors and natural data fluctuations, conditioned on the available covariate information in the dataset. We propose a simple yet efficient filtering procedure for eliminating potential errors, and establish theoretical guarantees for our method. We also contribute a new error detection benchmark involving 5 regression datasets with real-world numerical errors (for which the true values are also known). In this benchmark and additional simulation studies, our method identifies incorrect values with better precision/recall than other approaches.

**Keywords:**   aleatoric uncertainty, epistemic uncertainty, supervised machine learning

## 1 Introduction

Modern supervised machine learning has grown quite effective for most datasets thanks to development of highly-accurate models like random forests, gradient-boosting machines, and neural networks. Although it is generally assumed that the numeric responses (target values to predict) in the training data are accurate, this is often not the case in real-world datasets (Müller and Markert, 2019; Northcutt et al., 2021; Kang et al., 2022; Kuan and Mueller, 2022a). For classification data, many techniques have been proposed to address this issue by modifying training objectives or directly estimating which data is erroneous (Jiang et al., 2018; Zhang and Sabuncu, 2018; Song et al., 2022; Northcutt et al., 2021).

In this paper, we consider methods to identify similar erroneous values in regression datasets where the response is continuous[1]. Incorrect numeric values lurk in real-world data for many reasons including: measurement error (e.g. imperfect sensors), processing error (e.g. incorrect transformation of some values), recording error (e.g. data entry mistakes), or bad annotators (e.g. poorly trained data labelers) (Wang and Mueller, 2022; Kuan and Mueller, 2022a; Nettle, 2018; Agarwal et al., 2022). We are particularly interested in straightforward *model-agnostic* approaches that can utilize **any** type of regression model to identify the errors. These desiderata ensure our approach is applicable across diverse datasets in practice and can take advantage of state-of-the-art regressors (including future regression models not yet invented). Given a regression model, like Random Forest, Neural Network or Gradient Boosting Machine, we can use such model-agnostic approaches to estimate *all* erroneous responses in the dataset. Once identified, the datapoints with erroneous response may be filtered out from a dataset or fixed via external confirmation of the correct value to replace the incorrect one.

To help prioritize review of the most suspicious values, we consider a *veracity score* for each datapoint that reflects how likely a specific value is correct or not. Many prediction-based scores have been explored, such as residuals, likelihood values, and entropies (Northcutt et al., 2021; Kuan and Mueller, 2022a; Wang and Mueller, 2022; Wang and Jia, 2023; Thyagarajan et al., 2022; Ghorbani and Zou, 2019). While these methods are easy to implement and widely applicable, the uncertainties present in the observed data can impact prediction accuracy, consequently affecting both veracity scores and error detection. Two common types of uncertainties, epistemic and aleatoric, arise from a lack of observed data and intrinsic stochasticity in underlying relationships. Both types of uncertainties play a critical role in establishing the reliability of predictions.

In this paper, the introduced *veracity scores* incorporate both epistemic and aleatoric uncertainties. By accounting for these two types of uncertainties, an error detection procedure can more effectively distinguish between genuine anomalies and natural data fluctuations, ultimately resulting in more reliable identification of errors. Furthermore, we propose a simple yet efficient filtering procedure for eliminating potential errors. This algorithm automatically determines the number of errors to be removed and is compatible with any machine learning or statistical model. We introduce a comprehensive benchmark of datasets with naturally-occuring errors for which we also have corresponding ground truth values that can be used for evaluation. Results on this benchmark and extensive simulations illus-

---

1. Code to run our method: `https://github.com/cleanlab/cleanlab`
   Code to reproduce paper: `https://github.com/cleanlab/regression-label-error-benchmark`

trate the empirical effectiveness of our proposed approach to identify incorrect numerical responses in a dataset.

## 1.1 Related Work

A significant body of research has focused on identifying numerical outliers or anomalies that depend on contextual or conditional information. Song et al. (2007) introduced the concept of conditional outliers, which model outliers as influenced by a set of behavioral attributes (e.g., temperature) that are conditionally dependent on contextual factors (e.g., longitude and latitude). Valko et al. (2011) detected conditional anomalies using a training set of labeled examples, accounting for potential label noise. Tang et al. (2013) proposed an algorithm for detecting contextual outliers in categorical data based on attribute-value sets. Hong and Hauskrecht (2016) employed conditional probability to detect anomalies in clinical applications. Abedjan et al. (2016) conducted a comprehensive examination of multiple data cleaning tools, revealing that no single tool is universally dominant across all types of errors. Mahdavi et al. (2019) introduced Raha, a configuration-free error detection system that operates without the need for user-provided data constraints or parameters. Li et al. (2021) explored how data cleaning affects downstream machine learning models. Dasu and Loh (2012) proposed a statistical distortion measure and developed a versatile framework for analyzing and evaluating various cleaning strategies. For a comprehensive understanding on data cleaning problem, we recommend the monograph by Ilyas and Chu (2019). However, the methods proposed in these papers are model-specific and not universally applicable to all regression models, thereby limiting their utility in real-world data analysis involving complex data structures.

The random sample consensus (RANSAC) method proposed by Fischler and Bolles (1981) is a model-agnostic approach to error detection that iteratively identifies subsets of datapoints that are not well-fitted by a trained regression model. In contrast to RANSAC, our method effectively accounts for uncertainty in predictions from the regressor, which is crucial for differentiating confidently incorrect values from those that are merely inaccurately predicted. Conformal inference (Vovk et al., 2005; Lei et al., 2018; Bates et al., 2023) provides a framework to estimate the confidence in predictions from an arbitrary regressor, but we show here its direct application fares poorly when some data values are contaminated by noise.

## 2 Methods

### 2.1 Veracity scores

We consider a standard regression setting with covariates $X$ and a numerical response $Y$. We assume that the covariates $X$ are clean, but there might be errors in $Y$. Our goal is to utilize any fitted regression model to help detect observations $Y_i$ where the recorded value in the dataset is actually incorrect (i.e., corrupted).

Our approach constructs a numeric *veracity score* for each datapoint $X_i$, which reflects how likely $Y_i$ is correctly measured (based on how typical its value is given all of the other available information). For response variables $Y$ that are categorical, the predictive likelihood/entropy-based scores proposed by Kuan and Mueller (2022a) have demonstrated

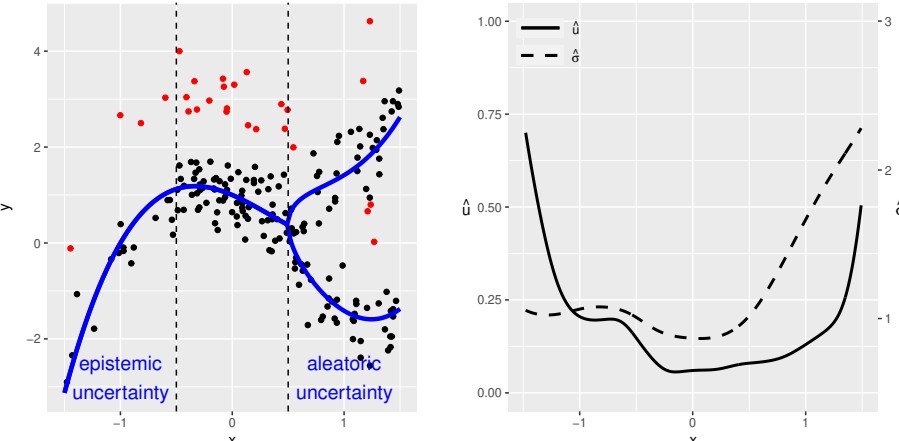

Figure 1: Left panel: Synthetic data with non-uniform epistemic and aleatoric uncertainties. 10% of the data points are set to be erroneous with a mean shift of 2, indicated in red.
Right pane: Estimated $\hat{u}(x)$ and $\hat{\sigma}(x)$, representing the quantification of epistemic and aleatoric uncertainties.

effective performance for identifying erroneous class labels via arbitrary classification models. Unlike standard classifiers, a typical regression model does not directly estimate the full conditional distribution of continuous response $Y$ (most models simply output point estimates). Thus a *model-agnostic* method (that can use **any** regression model) to detect errors in numerical data cannot employ analogous likelihood/entropy measures.

Throughout, all references to residuals and other prediction-based estimates (e.g. uncertainties) are assumed to be *out-of-sample*, i.e. produced for $X_i$ from a copy of the regression model that was never fit to this datapoint. Out-of-sample predictions can be obtained for an entire dataset through $K$-fold cross-validation, and are important to ensure less biased estimates for our *veracity scores* that are subject to less overfitting.

**Motivation:** For continuous response, the residual $\hat{S}_r(X_i, Y_i) = |Y_i - \hat{f}(X_i)|$ is a straightforward choice of score, where $\hat{f}$ represents the estimated regression function. Ideally, when the underlying relationship $f$ is relatively simple and $\hat{f}(x)$ is a well-fitted regressor, datapoints with abnormally large $\hat{S}_r(X_i, Y_i)$ values are likely to be anomalous values that warrant suspicion, provided the uncertainty (noise level in the model) is homoscedastic. However, this is no longer the case when the uncertainty is not homoscedastic.

Complexities of real-world data analysis make error detection more challenging, for instance non-uniform epistemic or aleatoric uncertainty due to lack of observations or heteroscedasticity. In the context of prediction, epistemic uncertainty results from a scarcity of observed data that is similar to a particular $X$, whose associated $Y$ value is thus hard to guess. On the other hand, aleatoric uncertainty results from inherent randomness in the underlying relationship between $X$ and $Y$ that cannot be reduced with additional data of the same covariates (but could by enriching the dataset with additional covariates). Figure

1 illustrates these two types of uncertainties: the epistemic uncertainty is large at a data-point $x$ with few nearby datapoints, while the aleatoric uncertainty is large at $x$ when the true underlying $Y|X = x$ is dispersed (e.g. a bimodal distribution).

After fitting a regression model in an expert manner, there are generally three reasons a residual $\hat{S}_r(X_i, Y_i)$ might be large:

- $Y_i$ was incorrectly measured (i.e. a data error).

- The estimation quality of $\hat{f}(x)$ is poor around $x = X_i$, e.g. due to large epistemic uncertainty (lack of sufficiently many observations similar to $X_i$).

- There is high aleatoric uncertainty, i.e. the underlying conditional distribution over target values, $Y|x = X_i$, is not concentrated around a single value.

Therefore, the residual score might be suboptimal for precise error detection due to false positive scores arising simply due to large uncertainty. We instead propose two veracity scores that rescale the residual in order to account for both epistemic and aleatoric uncertainties:

$$\hat{S}_a(x, y) = \frac{\hat{S}_r(x, y)}{\hat{u}(x) + \hat{\sigma}(x)} \qquad \hat{S}_g(x, y) = \frac{\hat{S}_r(x, y)}{\sqrt{\hat{u}(x)\hat{\sigma}(x)}}. \tag{1}$$

Here, epistemic uncertainty estimate $\hat{u}(x) := \sqrt{\widehat{\mathrm{Var}}(\hat{f}(x))}$ is the standard deviation of $\tilde{f}(x)$ over many regressors $\tilde{f}$ fit on bootstrap-resampled versions of the original data. Aleatoric uncertainty estimate $\hat{\sigma}(x) := \mathbb{E}(|\hat{f}(X) - Y| \mid X = x, \hat{f})$ is an estimate of the size of the regression error, produced by fitting a separate regressor to predict the residuals' size based on covariates $X$.

Our construction of $\hat{S}_a(x, y)$ and $\hat{S}_g(x, y)$ is a straightforward way to account for both epistemic and aleatoric uncertainties via their arithmetic or geometric mean. For datapoints where either uncertainty is abnormally large, the residuals are no longer reliable indicators. Thus presented with two datapoints whose $Y$ values deviate greatly from the predicted values (high residuals of say equal magnitude), we should be more suspicious of the datapoint whose corresponding prediction uncertainty is lower. In the synthetic data illustrated in Figure 1, the application of $\hat{S}_a$ successfully identified 19 out of 27 errors, demonstrating a markedly higher detection capability compared to the residual score, which identified only 6 errors.

## 2.2 Filtering procedure

A challenge arises in estimating regression model uncertainties from noisy data. Uncertainty estimates are based on the spread of $\hat{f}$ estimates, which are affected by the corrupted values in the dataset. Subsequent experiments in Section 4 show this same issue plagues the probabilistic estimates required for conformal inference. Here we propose a straightforward approach to mitigate this issue: simply filter some of the top most-confident errors from the dataset, and refit the regression model and its uncertainty estimates on the remaining less noisy data. For the best results, we can iterate this process until the noise has been sufficiently reduced. We use the following algorithm to iteratively filter potential errors in a dataset, $\mathbb{D}$. An overview flowchart is provided in Figure 2.

---

**Algorithm 1** Filtering procedure to reduce the amount of erroneous data

---

**Input:** Dataset $\mathbb{D}$; a regression model $\mathcal{A}$; the maximum proportion of corrupted data $K_{err}$.

1: Fit model $\mathcal{A}$ via $K$-fold cross-validation over the whole dataset, and compute veracity scores for each datapoint via out-of-sample predictions.
2: **for** $k = 1, 2, \ldots, K_{err}$ **do**
   a. Remove $k\%$ of the datapoints with the worst veracity scores. Denote the indices of removed datapoints as $\text{IND}_k$.
   b. Re-fit the model $\mathcal{A}$ with the remaining data (again via $K$-fold cross-validation) and denote the estimated regression function $\hat{f}$. Calculate the out-of-sample $R^2$ performance of the resulting predictions: $1 - \sum_{i \in \mathbb{D}} (y_i - \hat{f}(x_i))^2 / \sum_{i \in \mathbb{D}} (y_i - \bar{y})^2$ over the *entire* dataset,
   where $\bar{y} = \dfrac{1}{|\mathbb{D}|} \sum_{i \in \mathbb{D}} y_i$.
3: **end for**
4: Select the $k^*$ that produces the largest $R^2$ among $k = 1, \ldots, K_{err}$.

**Output:** Estimated corruption proportion $k^*$, indices of filtered data $\text{IND}_{k^*}$.

---

In Algorithm 1, the dataset $\mathbb{D} = (\boldsymbol{x}_i, y_i)_{i=1}^n$ imposes no restrictions on the covariates, allowing for numeric, text, images, or multimodal random variables in $\boldsymbol{x}_i$. The model $\mathcal{A}$ can be any parametric or non-parametric statistical regression model, or a machine learning model such as gradient boosting, random forest, or neural network. $K_{err}$ represents the maximum proportion of erroneous values that the user believes may be present in $\mathbb{D}$, generally not expected to exceed 20%. To reduce computation time, the grid search over $k \leq K_{err}$ can be replaced by a binary search or a coarse-then-fine grid.

In this algorithm, we use the (out-of-sample) $R^2$ metric as the criterion to assess the performance of the current removal process. It is crucial to evaluate the $R^2$ metric on the entire dataset $\mathbb{D}$, rather than only on the remaining data. Evaluating it solely on the remaining data would cause the $R^2$ value to increase continuously as more data points are removed. Although the complete dataset $\mathbb{D}$ contains errors, its corresponding $R^2$ value should improve if $\hat{f}$ is trained on a dataset with fewer errors. Conversely, a smaller sample size may lead to poorer performance of model $\mathcal{A}$ and a decrease in the corresponding $R^2$ value evaluated on $\mathbb{D}$, especially if we have started removing data that has no errors. Like RANSAC (Fischler and Bolles, 1981), this approach iteratively discards data and re-fits $\hat{f}$, but each iteration in our approach utilizes the veracity scores.

For the computational complexity of the proposed method, if we use $T$ to denote the amount of computation time needed to fit the baseline regression model once, the proposed method requires multiple fittings depending on the number of bootstrap $B$ and the number of filtering steps $F$. For each filtering step, we need $T$ time to train the model, an additional $T$ time to obtain the aleatoric uncertainty and $B*T$ time to obtain the epistemic uncertainty. In total, the computation complexity of our proposed algorithm is $F * (2 + B) * T$. In our numerical experiments, the resulted $F$ usually takes values 4 or 5. Although the Bootstrap method may be computationally intensive in some scenarios, its primary function is to assess data uncertainty and leverage this understanding to improve the estimation of residuals.

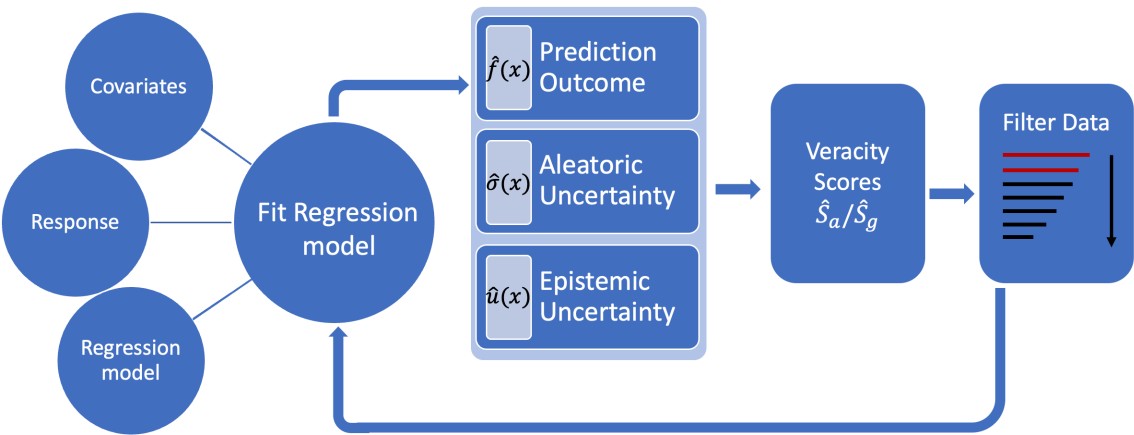

Figure 2: Flowchart of our proposed algorithm.

The cost for high-dimensional regressors is reflected in $T$. It is advisable to employ regressors that are effective in such settings. This adaptability is a key advantage of our proposed model-agnostic approach, as it allows for the selection of the most appropriate regressor tailored to the specific data characteristics.

## 3 Theoretical Analysis

This section analyzes when our algorithms can provably detect corrupted values in the dataset. We first provide a sufficient condition under which with probability exceeding 50%: the residual for a data point with a corrupted target value is greater than the residual at an uncorrupted target value. We then prove that with uncertainties in the data, our proposed scores are more likely to correctly detect erroneous data compared to basic residual scores.

We use $(X_i, Y_i)$ to denote a benign datapoint (whose $Y$-value is correct), where its distribution is given by $Y_i = f(X_i) + \epsilon_i(X_i)$, with $f(\cdot)$ denoting the true regression function and $\epsilon_i(X_i)$ denoting the traditional regression noise. On the other hand, an erroneous datapoint is represented as $(X_i', Y_i')$, with distribution $Y_i' = f(X_i') + \epsilon_i(X_i') + \epsilon_i^*(X_i')$, incorporating an additional corruption error: $\epsilon^*(X_i')$. In the scenario where the regression function is known, the $\hat{S}_r$ becomes: $S_r(X_i, Y_i) = |\epsilon_i(X_i)|$ for benign data $(X_i, Y_i)$, and $S_r(X_i', Y_i') = |\epsilon_i(X_i') + \epsilon_i(X_i')^*|$ for erroneous data $(X_i', Y_i')$. Let $F_x$ and $G_x$ represent the cumulative distribution functions (CDF) of $|\epsilon(X)|$ and $|\epsilon(X) + \epsilon^*(X)|$ at $X = x$, respectively, where $\epsilon(x)$ and $\epsilon^*(x)$ are prototypes error functions of $\epsilon_i(x)$ and $\epsilon_i^*(x)$, and $\epsilon_i(x)$ and $\epsilon_i^*(x)$ are independent and identically distributed (i.i.d.) samples from $\epsilon(x)$ and $\epsilon^*(x)$.

**Theorem 1** *Assume* $\mathbf{E}(\epsilon(X)|X) = 0$ *and is unimodal at* 0, $\epsilon(X)$ *and* $\epsilon^*(X)$ *are independent. If* $|\epsilon(X_i') + \epsilon^*(X_i')|$ *stochastically dominates* $|\epsilon(X_i)|$ *in the third order, that is,*

- $\int_{-\infty}^{x} \left[ \int_{-\infty}^{z} \{ F_{X_i}(t) - G_{X_i'}(t) \} \mathrm{d}t \right] \mathrm{d}z \geq 0$ *for all* $x$ *and*

- $\int_{\mathbb{R}} x \mathrm{d}G_{X_i'}(x) \geq \int_{\mathbb{R}} x \mathrm{d}F_{X_i}(x)$.

*Then,* $\mathbb{P}(S_r(X_i, Y_i) < S_r(X_i', Y_i')) \geq 1/2.$

Theorem 1 provides a sufficient condition ensuring that the probability of $S_r(X_i, Y_i) < S_r(X_i', Y_i')$ exceeds $1/2$. This implies that when the disparity between corrupted and clean target values is relatively large, the residual score can be effective for error detection. Third-order stochastic dominance is relatively weak and can be derived from first and second-order stochastic dominance. The subsequent corollary examines the case where the standard regression noise follows a Gaussian distribution, and the additional error corruption is a point mass at $a$. In that case, we have $F_{X_i}(t) - G_{X_i'}(t) \geq 0$, which implies that $|\epsilon(x') + \epsilon^*(x')|$ stochastically dominates $|\epsilon(x)|$ in the first order.

**Corollary 2** *If $\epsilon(x) \sim N(0, 1)$, $\epsilon^*(x) = a$ for all $x$. Then $\mathbb{P}(S_r(X_i, Y_i) < S_r(X_i', Y_i')) > 1/2$ for all $a \neq 0$ and $\mathbb{P}(S_r(X_i, Y_i) < S_r(X_i', Y_i')) \to 1$ exponentially as $a \to \infty$.*

When the estimated regression function $\hat{f}$ is consistent, $\hat{S}_r$ is asymptotically equivalent to the oracle case. The subsequent corollary directly follows from Theorem 1.

**Corollary 3** *Denote $\hat{f}$ the estimator of $f$ and $\hat{S}_r(X_i, Y_i) := |\hat{f}(X_i) - Y_i|$ the estimated residual scores. If $\|\hat{f} - f\|_\infty \xrightarrow{p} 0$, and $G_{X_i'}$ or $F_{X_i}$ is absolutely continuous, then $\mathbb{P}(\hat{S}_r(X_i, Y_i) < \hat{S}_r(X_i', Y_i')) = \mathbb{P}(S_r(X_i, Y_i) < S_r(X_i', Y_i')) + o(1).$*

The following theorem illustrates the conditions under which our proposed scores outperform the residual-based approach.

**Theorem 4** *Let $\hat{S}_a(X_i, Y_i)$ and $\hat{S}_g(X_i, Y_i)$ be the proposed veracity scores defined in 1,*

- *If $\hat{u}(X_i) + \hat{\sigma}(X_i) \geq \hat{u}(X_i') + \hat{\sigma}(X_i')$, $\mathbb{P}(\hat{S}_a(X_i, Y_i) < \hat{S}_a(X_i', Y_i')) \geq \mathbb{P}(\hat{S}_r(X_i, Y_i) < \hat{S}_r(X_i', Y_i')).$*

- *If $\hat{u}(X_i)\hat{\sigma}(X_i) \geq \hat{u}(X_i')\hat{\sigma}(X_i')$, $\mathbb{P}(\hat{S}_g(X_i, Y_i) < \hat{S}_g(X_i', Y_i')) \geq \mathbb{P}(\hat{S}_r(X_i, Y_i) < \hat{S}_r(X_i', Y_i')).$*

If $\hat{u}(X_i) > \hat{u}(X_i')$, that is, the bootstrap variance at $\hat{f}(X_i)$ is larger than that at $\hat{f}(X_i')$, it indicates greater epistemic uncertainty for $(X_i, Y_i)$. Similarly, if the variance of the regression error at $\hat{f}(X_i)$ exceeds that at $\hat{f}(X_i')$, there is higher aleatoric uncertainty for $(X_i, Y_i)$. In both instances, the residual might offer misleading information when assessing whether $Y_i$ is corrupted or not. Theorem 4 suggests that, in the presence of both epistemic and aleatoric uncertainties in the data, our proposed scores demonstrate superior performance compared to the residual.

## 4 Simulation Study

Here we present two experiments using diverse simulated datasets to evaluate the empirical performance of our proposed veracity scores as well as the filtering procedure. Two underlying settings are considered:

- **Setting 1: Non-parametric Regression with Epistemic/Aleatoric Uncertainty:** The covariates are i.i.d. from $\boldsymbol{x}_i = (x_{i1}, \dots, x_{i5}) \in \mathbb{R}^5$ with

$$x_{ij} \sim 0.1\text{Unif}(-1.5, -0.5) + 0.9\text{Unif}(-0.5, 1.5)$$

for $j = 1, \ldots, 5$. The *true* responses are generated by

$$y_i \sim 0.5N\left\{f(x_{i1}) - g(x_{i1}), 0.5\right\} + 0.5N\left\{f(x_{i1}) + g(x_{i1}), 0.5\right\},$$

where $f(x) = (x-1)^2(x+1)$, $g(x) = 2\sqrt{x - 0.5}\mathbb{1}(x \geq 0.5)$.

- **Setting 2: 5-D Linear Regression:** The *true* responses are generated by $y_i = \boldsymbol{\beta}^T \boldsymbol{x}_i + \epsilon_i$, where $\boldsymbol{x}_i \in \mathbb{R}^5$ and each coordinate of $\boldsymbol{x}_i$ is generated from $\mathrm{Unif}(-1.5, 1.5)$. The regression coefficients in $\boldsymbol{\beta}$ is set to $-1$ and $1$ with random signs and the regression error are i.i.d. $N(0, 0.5)$.

For both settings, the corrupted data is set to be $y_i^* = y_i + a$, that is, a point mass at $a$ with different corruption strength $a = -3, -2, -1, 1, 2, 3$. In the simulation study, the fraction of corrupted data is set to be 10% for all contaminated datasets.

Inspired by Lei and Wasserman (2014), Setting 1 involves a dataset that introduces both epistemic and aleatoric uncertainty. For each coordinate of $\boldsymbol{x}_i$, 90% of the $x_{ij}$ values are from $\mathrm{Unif}[-0.5, 1.5]$, while only 10% of the $x_{ij}$ values are from $\mathrm{Unif}[-1.5, -0.5]$. Consequently, the epistemic uncertainty for $x_{ij} \in [-1.5, -0.5]$ is larger due to insufficient observations for these $x_{ij}$. It is important to note that the response $y_i$ depends only on the first coordinate of $\boldsymbol{x}_i$, and the aleatoric uncertainty for those $x_{i1} \in [0.5, 1.5]$ is larger since $\mathbb{P}(y_i|\boldsymbol{x}_i)$ is bimodal. Figure 1 illustrates the observed $y_i$ with respect to the first coordinate of $\boldsymbol{x}_i$. Setting 2 is a simpler linear regression setting adapted from Hu and Lei (2020), where the residual should perform best as the model is simple and no additional uncertainty is involved. We consider settings in which we have clean training data and evaluate the error detection performance of methods in additional test data (no filtering needed), as well as settings where the entire dataset contains errors.

Our simulation study focuses on comparing our proposed veracity scores ($\hat{S}_a$ and $\hat{S}_g$) against the residual score $\hat{S}_r$, in order to investigate the empirical effect of additionally taking the regression uncertainties into account. In the Appendix C, we compare many alternative veracity scores against the residual score $\hat{S}_r$ over a diverse set of real datasets, and find that none of these alternatives is able to consistently outperform the residual score (making $\hat{S}_r$ a worthy baseline). Even though we know the underlying relationship in these simulations, we nonetheless fit a variety of popular regressors that are often used in practice: Random Forest (RF) (Breiman, 2001) and Gradient Boosting with LightGBM (LGBM) (Ke et al., 2017). All regression models fit in this paper (including the weighted ensemble) were implemented via the autogluon AutoML package (Erickson et al., 2020), which automatically provides good hyperparameter settings and manages the training of each model.

### 4.1 Conformal inference using the proposed scores

The conformal method has emerged as a leading tool for statistical inference, offering a generic approach to creating distribution-free prediction sets. Offering a $p$-value to test the conformity of data in the testing set, conformal inference is a natural procedure for outlier detection, as described by (Bates et al., 2023). However, the efficacy of the conformal method hinges on the exchangeability of the data; clean training and calibration sets are required (Bates et al., 2023). Such ideal conditions often elude real-world scenarios

where erroneous data is prevalent. Here we study the performance of conformal outlier detection against our proposed score for detecting corrupted values (particularly in settings where corrupted values are present in the training and calibration sets). We also consider alternative scores inspired by the conformal inference literature.

We first examine the performance of our proposed veracity scores in conformal inference. To reduce the computational burden associated with grid search, we utilize the splitting conformal method, which is widely adopted due to its efficiency (Chernozhukov et al., 2021; Bates et al., 2023). The splitting conformal method requires a training set to fit the model, a calibration set to evaluate the rank of the scores, and a testing set to assess performance. For each setting, the training and calibration sets are generated based on the aforementioned settings without errors. For the testing set, 10% of the data are designated as errors with a corruption strength $a = -3, -2, -1, 1, 2, 3$, while the remaining 90% are benign datapoints, having the same distributions as those in the training and calibration sets. For each $(X_i, Y_i)$ in the testing set, the conformal inference methodology enables us to obtain a $p$-value for the null hypothesis test $\mathcal{H}_{0,i} : X_i \sim P_0$ (Bates et al., 2023), where $P_0$ represents the distribution of the benign data. The error detection problem is then transformed into a multiple testing problem, and we can apply the Benjamini-Hochberg (BH) procedure to control the false discovery rate (FDR). This entire procedure is demonstrated in Algorithm 2 and Figure .

For each setting, we conduct 50 Monte-Carlo runs to mitigate the randomness that may occur in a single simulation. We use $\hat{S}_r$, $\hat{S}_a$ and $\hat{S}_g$ as the conformal scores $\hat{s}(x, y)$ in Algorithm 2 and the sample size $n = 200$ is the same for $\mathbb{D}^{\text{train}}$, $\mathbb{D}^{\text{cal}}$, and $\mathbb{D}^{\text{test}}$. For each run, we calculate the corresponding *False Discovery Rate* (FDR), the proportion of benign data among the test points incorrectly reported as errors, and the *Power*, the proportion of errors in the testing set correctly identified as errors. For each setting, two scenarios are considered, the first scenario is the typical conformal scenario where the training and calibration sets have no errors; while for the second scenario, the training and calibration sets are also contaminated and the errors proportion is the same to the testing set.

Table 1 presents the average FDR and Power for each setting where the training and calibration sets are clean. For Setting 1, which contains epistemic and aleatoric uncertainty, our proposed scores outperform the residual scores in both FDR and Power. For Setting 2, where the residual scores are expected to perform well, our proposed scores perform very closely to the residual scores and even surpass them in some cases. Table 2 shows that the conformal method fails when the training and calibration sets are contaminated. This is

---

**Algorithm 2** Conformal Outlier Detection

**Input:** Training set $\mathbb{D}^{\text{train}}$, calibration set $\mathbb{D}^{\text{cal}}$, and testing set $\mathbb{D}^{\text{test}}$; a model $\mathcal{A}$; a conformal score $s(x, y)$; a target FDR level $\alpha$.

1: Based on $\mathbb{D}^{\text{train}}$, obtain the estimated score $\hat{s}(X, Y)$.
2: Evaluate the scores $\{\hat{s}_i = \hat{s}(X_i, Y_i)\}_{i=1}^{\mathbb{D}^{\text{cal}}}$ for all datapoint in the calibration set, and denote the empirical CDF of $\{\hat{s}_i\}_{i=1}^n$ by $\hat{F}_{\hat{s}}$.
3: For each data point $(X_i, Y_i) \in \mathbb{D}^{\text{test}}$, get the conformal $p$-value $\hat{u}_i = (\hat{F}_{\hat{s}} \circ \hat{s})(X_i, Y_i)$.
4: Based on $\{\hat{u}_i\}_{i \in \mathbb{D}^{\text{test}}}$, apply BH procedure to determine which datapoint should be removed.

**Output:** Indices of outliers, i.e. datapoints expected to be erroneous.

---

Table 1: Average FDR and Power (with standard deviations in parentheses) for detecting corrupted values via conformal inference in different settings. In each setting, detection is based on a Random Forest regressor **trained on uncorrupted target values** (clean data available during training). The target FDR is 10% in all cases.

| | | Setting 1 | | | | | |
|---|---|---|---|---|---|---|---|
| corruption strength | | -3 | -2 | -1 | 1 | 2 | 3 |
| FDR | $\hat{S}_r$ | 0.16(0.16) | 0.30(0.33) | 0.44(0.43) | 0.37(0.44) | 0.19(0.30) | 0.16(0.16) |
| | $\hat{S}_a$ | 0.13(0.09) | 0.16(0.13) | 0.35(0.39) | 0.24(0.34) | 0.14(0.14) | 0.13(0.11) |
| | $\hat{S}_g$ | 0.12(0.09) | 0.18(0.15) | 0.36(0.39) | 0.34(0.40) | 0.14(0.15) | 0.13(0.09) |
| Power | $\hat{S}_r$ | 0.31(0.18) | 0.09(0.07) | 0.04(0.06) | 0.02(0.04) | 0.08(0.07) | 0.30(0.18) |
| | $\hat{S}_a$ | 0.71(0.13) | 0.41(0.22) | 0.06(0.07) | 0.06(0.06) | 0.38(0.17) | 0.72(0.11) |
| | $\hat{S}_g$ | 0.71(0.14) | 0.38(0.20) | 0.06(0.07) | 0.05(0.05) | 0.36(0.17) | 0.73(0.11) |
| | | Setting 2 | | | | | |
| FDR | $\hat{S}_r$ | 0.13(0.09) | 0.15(0.16) | 0.31(0.41) | 0.41(0.42) | 0.20(0.16) | 0.11(0.08) |
| | $\hat{S}_a$ | 0.13(0.09) | 0.15(0.16) | 0.34(0.38) | 0.27(0.33) | 0.20(0.17) | 0.13(0.09) |
| | $\hat{S}_g$ | 0.13(0.08) | 0.13(0.15) | 0.36(0.39) | 0.29(0.33) | 0.20(0.18) | 0.13(0.09) |
| Power | $\hat{S}_r$ | 0.77(0.14) | 0.24(0.15) | 0.03(0.05) | 0.04(0.05) | 0.29(0.16) | 0.81(0.15) |
| | $\hat{S}_a$ | 0.77(0.15) | 0.33(0.15) | 0.06(0.06) | 0.05(0.05) | 0.37(0.16) | 0.79(0.15) |
| | $\hat{S}_g$ | 0.77(0.15) | 0.32(0.15) | 0.05(0.06) | 0.04(0.04) | 0.37(0.16) | 0.80(0.15) |

because the validity of conformal inference crucially relies on the exchangeability (or some variant of exchangeability) between the calibration set and the future observation, thus if the calibration set is contaminated, the conformal prediction set will have biased coverage. Note that in the scenario where training, calibration and testing set are all equally noisy, this can be equivalently viewed as the performance for identifying errors in a given dataset.

Conformal inference often relies on scores based on conditional density or distribution functions. Chernozhukov et al. (2021) utilize an adjusted conditional distribution function as the conformity score to achieve optimal prediction intervals. Izbicki et al. (2020) use the distribution of the conditional density as the conformity score, demonstrating that its corresponding HPD-split conformal prediction sets have the smallest Lebesgue measure asymptotically. Figure 3 evaluates how well these alternative scores are able to detect corrupted values, revealing that our proposed scores remain more effective. Estimating conditional density or distribution functions also becomes challenging in settings with high-dimensional predictors, whereas our proposed scores can easily adapt to any high-dimensional regression model.

## 4.2 Filtering procedure

In this subsection, we examine the numerical performance of our proposed filtering procedure. For each setting in each Monte-Carlo run, we have $n = 200$ data points with 10% errors and corruption strength $a$. Given error detection can be viewed as an infor-

Table 2: Average FDR and Power (with standard deviations in parentheses) for detecting corrupted values via conformal inference in different settings. In each setting, detection is based on a Random Forest regressor **trained on 10% contaminated data**. The target FDR is 10% in all cases.

| | | Setting 1 | | | | | |
|---|---|---|---|---|---|---|---|
| corruption strength | | -3 | -2 | -1 | 1 | 2 | 3 |
| FDR | $\hat{S}_r$ | 0.03(0.14) | 0.13(0.28) | 0.38(0.44) | 0.48(0.48) | 0.14(0.31) | 0.08(0.22) |
| | $\hat{S}_a$ | 0.01(0.07) | 0.03(0.15) | 0.18(0.33) | 0.29(0.40) | 0.04(0.13) | 0.00(0.00) |
| | $\hat{S}_g$ | 0.01(0.04) | 0.03(0.15) | 0.23(0.38) | 0.29(0.41) | 0.03(0.09) | 0.00(0.00) |
| Power | $\hat{S}_r$ | 0.05(0.06) | 0.04(0.06) | 0.01(0.03) | 0.00(0.01) | 0.03(0.05) | 0.05(0.07) |
| | $\hat{S}_a$ | 0.05(0.07) | 0.05(0.07) | 0.03(0.04) | 0.02(0.03) | 0.03(0.05) | 0.05(0.06) |
| | $\hat{S}_g$ | 0.05(0.06) | 0.05(0.06) | 0.02(0.03) | 0.02(0.04) | 0.04(0.06) | 0.06(0.06) |

mation retrieval problem, we follow Kuan and Mueller (2022a) and use the Area Under the Precision-Recall Curve (AUPRC) metric to evaluate various veracity scores. AUPRC quantifies how well these scores are able to rank erroneous datapoints above those with correct values, which is essential to effectively handle errors in practice.

Table 4 presents the average AUPRC based on the original dataset, proportion of corruptions removed, proportion of corruptions in the remaining data, and AUPRC based on the remaining data for 50 Monte-Carlo runs. We observe that in both Setting 1 and Setting 2, the corruption proportions in the remaining data decrease as the corruption strength increases, and the AUPRC improves after running our removal algorithm. Furthermore, the removed proportion is very close to the true corruption proportion in the original dataset. In Setting 1, which includes epistemic and aleatoric uncertainty, our proposed scores $\hat{S}_a$ and $\hat{S}_g$ outperform the residual score in AUPRC across all scenarios. In Setting 2, where the underlying uncertainty should be relatively uniform, our proposed scores perform similarly to the residual score.

## 5 Benchmark with Real Data and Real Errors

Here, we evaluate the performance of our proposed methods using five publicly available datasets. For each dataset, we have an observed target value that we use for fitting regression models and computing veracity scores and other estimates. For evaluation, we also have a true target value available in each dataset (not made available to any of our estimation procedures). For instance, in the **Air CO** air quality dataset, the observed target values stem from an inferior sensor device, whereas the true target values stem from a much high-quality sensor placed in the same locations. Detailed information regarding these datasets can be found in Section B of the Supplement.

The proportion of actual errors lurking in each dataset varies. We first evaluate the error-detection performance of our proposed scores compared to the residuals in settings where the regression model is trained on clean data with uncorrupted target values. We consider

Table 3: AUROC/AUPRC for detecting corrupted values in Setting 1 via different scoring methods. Each reported value is an average over 50 Monte Carlo replicate runs with different data. CZK and HPD are conformal-based scores proposed by Chernozhukov et al. (2021) and Izbicki et al. (2020), applied with two methods for estimating conditional density/distribution functions: random forest (Pospisil and Lee, 2018, RF) and FlexCode (Izbicki and Lee, 2017, FLEX). Our proposed scores are computed using the same Random Forest regressor as before.

| | corruption strength | -3 | -2 | -1 | 1 | 2 | 3 |
|---|---|---|---|---|---|---|---|
| AUROC | $\hat{S}_r$ | 0.97 | 0.87 | 0.67 | 0.71 | 0.87 | 0.97 |
| | $\hat{S}_a$ | 0.95 | 0.90 | 0.72 | 0.76 | 0.88 | 0.94 |
| | $\hat{S}_g$ | 0.95 | 0.90 | 0.72 | 0.76 | 0.88 | 0.94 |
| | CZK-RF | 0.93 | 0.84 | 0.51 | 0.70 | 0.79 | 0.94 |
| | CZK-FLEX | 0.96 | 0.86 | 0.54 | 0.73 | 0.83 | 0.99 |
| | HPD-RF | 0.92 | 0.86 | 0.75 | 0.71 | 0.88 | 0.89 |
| | HPD-FLEX | 0.94 | 0.85 | 0.75 | 0.66 | 0.84 | 0.90 |
| AUPRC | $\hat{S}_r$ | 0.83 | 0.49 | 0.20 | 0.22 | 0.53 | 0.84 |
| | $\hat{S}_a$ | 0.88 | 0.80 | 0.38 | 0.45 | 0.78 | 0.89 |
| | $\hat{S}_g$ | 0.88 | 0.79 | 0.38 | 0.44 | 0.78 | 0.89 |
| | CZK-RF | 0.63 | 0.31 | 0.11 | 0.21 | 0.36 | 0.58 |
| | CZK-FLEX | 0.80 | 0.40 | 0.15 | 0.21 | 0.36 | 0.94 |
| | HPD-RF | 0.45 | 0.31 | 0.30 | 0.18 | 0.37 | 0.34 |
| | HPD-FLEX | 0.73 | 0.38 | 0.24 | 0.14 | 0.52 | 0.42 |

four types of regression models: Gradient Boosting with LightGBM (Ke et al., 2017), Feedforward Neural Network (NN) (Gurney, 1997), Random Forest (Breiman, 2001), and a Weighted Ensemble of these models fit via Ensemble Selection (WE) (Caruana et al., 2004). Estimates are evaluated using four metrics popular in information retrieval applications: area under the receiver operating characteristic curve (AUROC), AUPRC, lift at $k$ (where $k$ is the true underlying number of errors in dataset), and lift at 100. Each metric evaluates how well a method is able to retrieve or rank the corrupted datapoints ahead of the benign data.

Table 5 shows the average improvement of our proposed scores compared to the residual scores. For example, the first four numbers in the first row represent $(\text{AUROC}(\hat{S}_a) - \text{AUROC}(\hat{S}_r))/\text{AUROC}(\hat{S}_r)$ corresponding to the LightGBM, Neural Network, Random Forest, and Weighted Ensemble models. A larger positive percentage indicates better performance of our proposed scores. Table 5 contains few negative values, implying our proposed

Table 4: Error-detection performance (AUPRC) under various data filtering methods. All scores are calculated via cross-validation with a LightGBM regression model. **AUPRC before** is achieved from scores based on the original dataset (no filtering); **AUPRC after** is achieved from scores based on the filtered dataset; **removed prop** is the proportion of data removed by our filter algorithm; **error prop** is the proportion of corrupted values remaining in the filtered dataset.

| corruption strength | | Setting 1 | | | | Setting 2 | | | |
|---|---|---|---|---|---|---|---|---|---|
| | | AUPRC before | removed prop | error prop | AUPRC after | AUPRC before | removed prop | error prop | AUPRC after |
| a=-3 | $\hat{S}_r$ | 0.60 | 11.74% | 5.13% | 0.66 | 0.84 | 14.64% | 2.62% | 0.94 |
| | $\hat{S}_a$ | 0.64 | 11.22% | 4.98% | 0.72 | 0.78 | 14.88% | 2.58% | 0.93 |
| | $\hat{S}_g$ | 0.62 | 9.60% | 5.62% | 0.71 | 0.78 | 15.36% | 2.50% | 0.93 |
| a=-2 | $\hat{S}_r$ | 0.33 | 12.34% | 7.55% | 0.34 | 0.63 | 12.92% | 4.43% | 0.70 |
| | $\hat{S}_a$ | 0.40 | 11.64% | 6.89% | 0.44 | 0.60 | 13.14% | 4.52% | 0.68 |
| | $\hat{S}_r$ | 0.37 | 11.74% | 7.01% | 0.43 | 0.60 | 13.18% | 4.54% | 0.66 |
| a=-1 | $\hat{S}_r$ | 0.15 | 11.26% | 9.37% | 0.16 | 0.25 | 11.38% | 8.01% | 0.28 |
| | $\hat{S}_a$ | 0.16 | 11.98% | 8.82% | 0.19 | 0.24 | 12.08% | 8.09% | 0.27 |
| | $\hat{S}_g$ | 0.16 | 10.66% | 9.14% | 0.18 | 0.24 | 12.00% | 8.08% | 0.27 |
| a=1 | $\hat{S}_r$ | 0.16 | 11.04% | 9.39% | 0.15 | 0.25 | 11.66% | 7.86% | 0.28 |
| | $\hat{S}_a$ | 0.21 | 12.72% | 8.29% | 0.20 | 0.24 | 10.92% | 8.15% | 0.27 |
| | $\hat{S}_g$ | 0.20 | 13.88% | 8.24% | 0.19 | 0.24 | 11.72% | 8.01% | 0.27 |
| a=2 | $\hat{S}_r$ | 0.33 | 11.18% | 7.01% | 0.31 | 0.59 | 12.48% | 4.71% | 0.71 |
| | $\hat{S}_a$ | 0.40 | 11.36% | 6.11% | 0.43 | 0.57 | 13.50% | 4.79% | 0.69 |
| | $\hat{S}_g$ | 0.38 | 11.76% | 6.19% | 0.40 | 0.55 | 12.56% | 5.07% | 0.66 |
| a=3 | $\hat{S}_r$ | 0.59 | 10.56% | 5.29% | 0.67 | 0.84 | 11.66% | 7.86% | 0.95 |
| | $\hat{S}_a$ | 0.62 | 10.04% | 5.24% | 0.70 | 0.79 | 10.92% | 8.15% | 0.92 |
| | $\hat{S}_g$ | 0.61 | 11.34% | 5.20% | 0.68 | 0.79 | 11.72% | 8.01% | 0.91 |

scores outperform residuals in the overwhelming majority of cases. These empirical results agree with our prior theoretical analysis – our scores consistently deliver better performance than the residuals for datasets with higher aleatoric or epistemic uncertainty. For simple datasets where the residual approach already performs effectively, our method does not noticeably improve error detection compared to this baseline. For datasets where the level of corruption is severe, our method greatly outperforms the baseline residuals approach (in part because our filtering procedure significantly enhances the robustness of the regression estimates in such settings).

One may find that improvements in Table 5 varies for different regression models. Since our method is designed to be model-agnostic, which means that it does not require a specific choice of regression model or training procedure. Of course the predictive accuracy of the model will influence the subsequent results. Thus, to use the method effectively, we recommend no change to the existing machine learning workflow – use whatever tricks and training procedures that will get you the most accurate model on your data. And then directly employ our method afterwards. This generality ensures our method will remain applicable in the future as novel regression algorithms are invented. We believe this is a critical property – methods that only work for say random forests would become obsolete

in the age of deep learning, and methods that only work for today's neural networks would also become obsolete when a better regression model is invented in the future.

Next, we compare our proposed filtering procedure with the RANSAC algorithm (Fischler and Bolles, 1981). When applying RANSAC, we used its default hyperparameter settings in the *scikit-learn* package. Here we separately run each of these data filtering procedures, and then compute three veracity scores $\hat{S}_a, \hat{S}_g, \hat{S}_r$ from the same type of model fit to the filtered data. Some values in the table are left blank because the RANSAC algorithm from the `scikit-learn` package can only handle numeric covariates, which excludes the "Stanford Politeness Wiki" dataset. Table 6 shows that, when the entire dataset may contain corrupted values, our proposed filtering procedure generally performs better than RANSAC, which tends to overestimate or underestimate the corruption proportions. Furthermore, our veracity score combined with our filtering procedure leads to the best overall error detection performance across these datasets.

The effectiveness of our proposed filtering procedure is further evaluated using the prediction error metric $\mathcal{E}_p$, defined as:

$$\mathcal{E}_p := \sum_{i=1}^{n} (\hat{y}_{-i} - y_i)^2,$$

where $\hat{y}_{-i}$ represents the leave-one-out prediction for $y_i$. As demonstrated in Table 7, there is a significant reduction in the prediction error $\mathcal{E}_p$ across all veracity scores ($\hat{S}_r$, $\hat{S}_a$, and $\hat{S}_g$) after the application of our filtering procedure. Furthermore, our proposed scores, $\hat{S}_a$ and $\hat{S}_g$, outperform the baseline residual in most datasets.

## 6 Discussion

For detecting erroneous numerical values in real-world data, this paper introduces novel veracity scores to quantify how likely each datapoint's $Y$-value has been corrupted. When we have a clean training dataset that is used to detect errors in subsequent test data, these veracity scores significantly outperform residuals alone, by properly accounting for epistemic and aleatoric uncertainties. When the entire dataset may contain corruptions, the uncertainty estimates degrade. For this setting, we introduce a filtering procedure that reduces the amount of corruption in the dataset. Such filtering helps us obtain better uncertainty estimates that result in more effective veracity scores for detecting erroneous values. We present a comprehensive benchmark of real-world regression datasets with naturally occurring erroneous values, over which our proposed approaches outperform other methods. All of our proposed approaches work with any regression model, which makes them widely applicable. Armed with our methods to detect corrupted data, data scientists will be able to produce more reliable models/insights out of noisy datasets.

As outlined in the theory section, when faced with higher aleatoric or epistemic uncertainty in the data, our scores consistently deliver better performance than the residuals. For datasets possessing a simple structure (without much aleatoric/epsitemic uncertainty) wherein the residuals already perform effectively, our method does not noticeably improve error detection. Furthermore, for datasets where the level of corruption is severe, our method greatly outperforms the baseline residuals approach, because our filtering procedure significantly enhances the robustness of the regression estimates in such settings.

In some real data, corruptions may be systematically biased, which no statistical procedure can detect, and thus our method will fail to outperform the baseline residuals approach in such cases. Outlier detection for datasets with systematically biased corruptions is an interesting problem and the framework in this paper is useful for further research.

Table 5: Percentage improvement of arithmetic or geometric scores vs. residual for detecting errors in each dataset (according to various evaluation metrics listed in the second column). Regression models are trained on clean data (uncorrupted target values), and then various scores are computed over the whole dataset using these models.

| data set | metric | $\hat{S}_a$ | | | | $\hat{S}_g$ | | | |
| --- | --- | --- | --- | --- | --- | --- | --- | --- | --- |
| | | LGBM | NN | RF | WE | LGBM | NN | RF | WE |
| Air CO | auroc | -0.37% | -0.40% | 2.57% | 1.48% | -1.17% | -3.25% | 2.92% | 0.64% |
| | auprc | 38.47% | 13.33% | 41.58% | 138.76% | 44.42% | -1.39% | 43.21% | 117.49% |
| | lift_at_num_errors | 23.48% | 10.53% | 15.76% | 64.20% | 24.35% | -3.51% | 18.79% | 55.56% |
| | lift_at_100 | 44.00% | 70.00% | 39.71% | 156.76% | 50.00% | 35.00% | 41.18% | 127.03% |
| metaphor | auroc | 3.10% | 1.20% | 5.05% | 4.62% | 3.21% | 3.04% | 6.56% | 7.72% |
| | auprc | 64.76% | 62.95% | 92.87% | 64.73% | 70.80% | 99.15% | 114.37% | 73.55% |
| | lift_at_num_errors | 21.95% | 39.13% | 39.76% | 49.47% | 25.61% | 43.48% | 55.42% | 56.84% |
| | lift_at_100 | 55.00% | 53.85% | 74.42% | 66.10% | 67.50% | 96.15% | 104.65% | 66.10% |
| stanford stack | auroc | 1.20% | 0.96% | 0.99% | 0.48% | 1.24% | 0.59% | 1.08% | 0.66% |
| | auprc | 11.53% | 13.30% | 10.77% | 1.76% | 11.72% | 10.26% | 11.31% | 2.21% |
| | lift_at_num_errors | 11.92% | 11.88% | 10.00% | 6.70% | 13.25% | 5.00% | 11.88% | 6.70% |
| | lift_at_100 | 6.38% | 9.89% | 6.38% | 0.00% | 6.38% | 8.79% | 6.38% | 0.00% |
| stanford wiki | auroc | 0.85% | -0.91% | 1.01% | 1.01% | 1.14% | -1.02% | 1.10% | 1.44% |
| | auprc | 13.54% | 5.29% | 8.65% | 8.76% | 14.99% | 5.16% | 8.61% | 10.12% |
| | lift_at_num_errors | 6.07% | 2.66% | 4.22% | 6.01% | 6.54% | 3.19% | 4.22% | 9.44% |
| | lift_at_100 | 14.94% | 6.98% | 6.38% | 5.26% | 14.94% | 5.81% | 6.38% | 5.26% |
| telomere | auroc | -0.34% | -0.08% | 0.15% | 0.00% | -1.39% | -1.33% | 0.15% | -0.28% |
| | auprc | 0.77% | -1.18% | 4.38% | 0.14% | -10.15% | -15.50% | 4.29% | -2.81% |
| | lift_at_num_errors | -3.41% | -3.89% | 8.37% | 0.22% | -15.61% | -20.14% | 7.14% | -6.87% |
| | lift_at_100 | 3.09% | 0.00% | 1.01% | 0.00% | 3.09% | -3.00% | 1.01% | 1.01% |

Table 6: Our proposed data filtering procedure compared with the RANSAC algorithm. Both approaches are applied with a LightGBM regressor. Our veracity score is computed after data filtering to assess final error-detection performance (AUROC/AUPRC).

| | | | Our proposed filtering procedure | | | | RANSAC in sklearn | | | |
| --- | --- | --- | --- | --- | --- | --- | --- | --- | --- | --- |
| | | original error% | removed% | error% after | AUROC | AUPRC | removed% | error% after | AUROC | AUPRC |
| Air CO | $\hat{S}_r$ | | 2.00% | 5.03% | 0.58 | 0.08 | | | 0.56 | 0.08 |
| | $\hat{S}_a$ | 5.13% | 7.01% | 4.89% | 0.58 | 0.07 | 2.11% | 4.90% | 0.55 | 0.08 |
| | $\hat{S}_g$ | | 6.00% | 4.98% | 0.54 | 0.06 | | | 0.54 | 0.07 |
| metaphor | $\hat{S}_r$ | | 5.03% | 6.33% | 0.93 | 0.09 | | | 0.64 | 0.10 |
| | $\hat{S}_a$ | 6.55% | 17.99% | 5.98% | 0.94 | 0.09 | 43.58% | 4.88% | 0.64 | 0.10 |
| | $\hat{S}_g$ | | 19.01% | 6.01% | 0.94 | 0.09 | | | 0.62 | 0.09 |
| Stanford stack | $\hat{S}_r$ | | 3.06% | 10.09% | 0.93 | 0.65 | | | 0.95 | 0.69 |
| | $\hat{S}_a$ | 11.86% | 5.01% | 8.17% | 0.94 | 0.68 | 72.37% | 0.44% | 0.93 | 0.65 |
| | $\hat{S}_g$ | | 4.03% | 8.79% | 0.94 | 0.73 | | | 0.94 | 0.67 |
| Stanford wiki | $\hat{S}_r$ | | 22.04% | 13.31% | 0.82 | 0.60 | | | | |
| | $\hat{S}_a$ | 22.96% | 24.03% | 12.55% | 0.82 | 0.64 | | | | |
| | $\hat{S}_g$ | | 10.07% | 17.22% | 0.82 | 0.64 | | | | |
| telomere | $\hat{S}_r$ | | 16.00% | 0.04% | 0.99 | 0.78 | | | 0.99 | 0.77 |
| | $\hat{S}_a$ | 4.66% | 18.00% | 0.07% | 0.99 | 0.83 | 0.62% | 4.18% | 0.99 | 0.80 |
| | $\hat{S}_g$ | | 22.00% | 0.10% | 0.97 | 0.73 | | | 0.97 | 0.72 |

Table 7: Comparison of Leave-One-Out prediction errors: original data vs. data processed with proposed filtering procedure with veracity scores ($\hat{S}_r$, $\hat{S}_a$, $\hat{S}_g$). Both approaches are applied with a LightGBM regressor. The values in columns marked with an asterisk (*) have been multiplied by 100 for visualization.

| | | Air CO* | Metaphor* | Stanford stack | Stanford wiki | Telomere* |
| --- | --- | --- | --- | --- | --- | --- |
| original prediction error | | 4.37 | 28.40 | 10.02 | 8.26 | 0.273 |
| prediction error after filtering | $\hat{S}_r$ | 2.40 | 12.26 | 3.52 | 5.57 | 0.139 |
| | $\hat{S}_a$ | 2.16 | 13.28 | 3.20 | 2.79 | 0.126 |
| | $\hat{S}_g$ | 2.76 | 12.88 | 2.52 | 3.19 | 0.143 |

## Appendix A. Proofs of theorems in Section 3

**Proof** [of Theorem 1] Note that

$$\mathbb{P}(S_r(X_i, Y_i) < S_r(X_i', Y_i')) = \iint_{x>y} \mathrm{d}G_{X_i'}(x)\mathrm{d}F_{X_i}(y) = \int_{-\infty}^{\infty} \int_{-\infty}^{x} \mathrm{d}F_{X_i}(y)\mathrm{d}G_{X_i'}(x)$$

$$= \int_{-\infty}^{\infty} F_{X_i}(x)\mathrm{d}G_{X_i'}(x).$$

If $G$ stochastically dominates $F$ in the third order, then $\mathbf{E}_G U(x) \geq \mathbf{E}_F U(x)$, for all nonde-creasing, concave utility functions $U$ that are positively skewed. Since $F_{X_i}(x)$ is the CDF of absolute value of the regression error, it is obviously nondecreasing and positively skewed. To see $F_{X_i}(x)$ is concave, note that

$$\frac{\mathrm{d}^2 F_{X_i}(x)}{\mathrm{d}x^2} = \frac{\mathrm{d}f_{X_i}(x)}{\mathrm{d}x} + \frac{\mathrm{d}f_{X_i}(-x)}{\mathrm{d}x} = 2\frac{\mathrm{d}f_{X_i}(x)}{\mathrm{d}x} \leq 0, \text{ for } x > 0,$$

where $f_{X_i}(x)$ is the density function of $F_{X_i}(x)$ and the last inequality is from $\mathbf{E}(\epsilon(x)) = 0$ and is unimodal at 0. Thus, $F_{X_i}(x)$ is a concave utility function and

$$\int_{-\infty}^{\infty} F_{X_i}(x)\mathrm{d}G_{X_i'}(x) \geq \int_{-\infty}^{\infty} F_{X_i}(x)\mathrm{d}F_{X_i}(x) = \int_0^1 x\mathrm{d}x = \frac{1}{2},$$

which complete the proof. ■

**Proof** [of Corollary 2] Under the assumption of Corollary 2, $\epsilon_i(X_i) \sim N(0,1)$, $\epsilon(X_i') + \epsilon^*(X_i') \sim N(a,1)$. Thus

$$\mathrm{d}G_{X_i'}(x) = \left\{ \frac{1}{\sqrt{2\pi}}e^{-\frac{(x-a)^2}{2}} + \frac{1}{\sqrt{2\pi}}e^{-\frac{(x+a)^2}{2}} \right\} \mathrm{d}x, \mathrm{d}F_{X_i}(y) = \frac{2}{\sqrt{2\pi}}e^{-\frac{y^2}{2}}\mathrm{d}y.$$

Denote

$$f(a) := \mathbb{P}(S_r(X_i, Y_i) < S_r(X_i', Y_i'))$$

$$= \int_0^{\infty} \int_0^x \left\{ \frac{1}{\sqrt{2\pi}}e^{-\frac{(x-a)^2}{2}} + \frac{1}{\sqrt{2\pi}}e^{-\frac{(x+a)^2}{2}} \right\} \frac{2}{\sqrt{2\pi}}e^{-\frac{y^2}{2}}\mathrm{d}x\mathrm{d}y$$

$$= \frac{1}{2}\sqrt{\frac{2}{\pi}} \int_0^{\infty} \left\{ e^{-\frac{(x-a)^2}{2}} + e^{-\frac{(x+a)^2}{2}} \right\} \mathrm{Erf}\left(\frac{x}{\sqrt{2}}\right)\mathrm{d}x,$$

where $\mathrm{Erf}(z) = 2\pi^{-1/2}\int_0^z e^{-t^2}\mathrm{d}t$ is the error function. Note that $f(0) = 1/2$, we hope to show

$$f(a) - f(0) = \frac{1}{2}\sqrt{\frac{2}{\pi}} \int_0^{\infty} \left\{ e^{-\frac{(x-a)^2}{2}} + e^{-\frac{(x+a)^2}{2}} - 2e^{-\frac{x^2}{2}} \right\} \mathrm{Erf}\left(\frac{x}{\sqrt{2}}\right)\mathrm{d}x \geq 0.$$

A change of variable leads to

$$\int_0^\infty \left\{ e^{-\frac{(x-a)^2}{2}} + e^{-\frac{(x+a)^2}{2}} - 2e^{-\frac{x^2}{2}} \right\} \operatorname{Erf}\left(\frac{x}{\sqrt{2}}\right) dx$$

$$= \int_0^\infty \left\{ e^{-\frac{(x-a)^2}{2}} + e^{-\frac{(x+a)^2}{2}} - 2e^{-\frac{x^2}{2}} \right\} \int_0^{\frac{x}{\sqrt{2}}} e^{-t^2} dt dx$$

$$= \frac{2}{\sqrt{\pi}} \int_0^\infty e^{-t^2} \int_{\sqrt{2t}}^\infty \left\{ e^{-\frac{(x-a)^2}{2}} + e^{-\frac{(x+a)^2}{2}} - 2e^{-\frac{x^2}{2}} \right\} dx dt$$

$$= \sqrt{2} \int_0^\infty e^{-t^2} \left\{ 2\operatorname{Erf}(t) - \operatorname{Erf}\left(t - \frac{a}{\sqrt{2}}\right) - \operatorname{Erf}\left(t + \frac{a}{\sqrt{2}}\right) \right\} dt$$

$$= \sqrt{2} \int_0^\infty e^{-t^2} \left\{ \int_{t-\frac{a}{\sqrt{2}}}^t e^{-u^2} du - \int_t^{t+\frac{a}{\sqrt{2}}} e^{-u^2} du \right\} dt.$$

Since $\int_{t-\frac{a}{\sqrt{2}}}^t e^{-u^2} du - \int_t^{t+\frac{a}{\sqrt{2}}} e^{-u^2} du$ is always positive due to the monotonicity of $e^{-u^2}$, which implies $f(a) - f(0) > 0$. Thus $\mathbb{P}(S_r(X_i, Y_i) < S_r(X_i', Y_i')) > 1/2$. Furthermore, note that $\lim_{a\to\infty} 2\operatorname{Erf}(t) - \operatorname{Erf}\left(t - a/\sqrt{2}\right) - \operatorname{Erf}\left(t + a/\sqrt{2}\right) \to 2\operatorname{Erf}(t)$ exponentially. Thus

$$\sqrt{2} \int_0^\infty e^{-t^2} \left\{ 2\operatorname{Erf}(t) - \operatorname{Erf}\left(t - \frac{a}{\sqrt{2}}\right) - \operatorname{Erf}\left(t + \frac{a}{\sqrt{2}}\right) \right\} dt$$

$$\to \sqrt{2} \int_0^\infty e^{-t^2} 2\operatorname{Erf}(t) dt = \sqrt{\frac{\pi}{2}},$$

which complete the proof. ∎

**Proof** [of Corollary 3] For all $\delta > 0$, note that

$$\mathbb{P}(\hat{S}_r(X_i, Y_i) < \hat{S}_r(X_i', Y_i'))$$
$$\geq \mathbb{P}(\hat{S}_r(X_i, Y_i) < \hat{S}_r(X_i', Y_i'), \|\hat{f} - f\|_\infty \leq \delta)$$
$$= \mathbb{P}(|f(X_i) - Y_i + \hat{f}(X_i) - f(X_i)| < |f(X_i') - Y_i' + \hat{f}(X_i') - f(X_i')|, \|\hat{f} - f\|_\infty \leq \delta)$$
$$\geq \mathbb{P}(|f(X_i) - Y_i| < |f(X_i') - Y_i'| - 2\delta, \|\hat{f} - f\|_\infty \leq \delta)$$
$$\geq \mathbb{P}(|f(X_i) - Y_i| < |f(X_i') - Y_i'| - 2\delta) - \mathbb{P}(\|\hat{f} - f\|_\infty > \delta).$$

For the first term in the right hand side of last equation,

$$\mathbb{P}(|f(X_i) - Y_i| < |f(X_i') - Y_i'| - 2\delta) = \iint_{x>y+2\delta} dG_{X_i'}(x) dF_{X_i}(y)$$

$$= \int \left[ \int_{x>y} - \int_{x\in(y,y+2\delta)} \right] dG_{X_i'}(x) dF_{X_i}(y).$$

If $G_{X_i'}$ or $F_{X_i}$ is absolutely continuous, then

$$\int \left[ \int_{x>y} - \int_{x\in(y,y+2\delta)} \right] dG_{X_i'}(x) dF_{X_i}(y) = o(1) \text{ as } \delta \to 0.$$

Under the assumption $\|\hat{f} - f\|_\infty \xrightarrow{p} 0$, $\mathbb{P}(\|\hat{f} - f\|_\infty > \delta) = o(1)$ for all $\delta > 0$, which complete the proof. ∎

**Proof** [of Theorem 4] The proof is straight forward since

$$
\begin{aligned}
&\mathbb{P}(\hat{S}_a(X_i, Y_i) < \hat{S}_a(X_i', Y_i')) \\
=&\mathbb{P}\left( \frac{\hat{S}_r(X_i', Y_i')}{\hat{u}(X_i') + \hat{\sigma}(X_i')} \leq \frac{\hat{S}_r(X_i, Y_i)}{\hat{u}(X_i) + \hat{\sigma}(X_i)} \right) = \mathbb{P}\left( \hat{S}_r(X_i', Y_i') \leq \frac{\hat{u}(X_i') + \hat{\sigma}(X_i')}{\hat{u}(X_i) + \hat{\sigma}(X_i)} \hat{S}_r(X_i, Y_i) \right) \\
\geq&\mathbb{P}\left( \hat{S}_r(X_i', Y_i') \leq \hat{S}_r(X_i, Y_i) \right) = \mathbb{P}(\hat{S}_r(X_i, Y_i) < \hat{S}_r(X_i', Y_i')).
\end{aligned}
$$

∎

## Appendix B. Benchmark Details

For each dataset, we have a given_label representing the noisily-measured response variable typically available in real-world datasets, and a true_label representing a higher fidelity approximation of the true $Y$ value one wishes to measure. The true_label would be unavailable for most datasets in practice and is here solely used for evaluation of different error detection methods. To determine *which* datapoints should be considered truly erroneous in a particular dataset, we conducted a histogram and Gaussian kernel density analysis of true_label - given_label in each dataset, and identified where these deviations became atypically large. Below we list some additional details about each dataset.

**Air Quality dataset**: This benchmark dataset is a subset of data provided by the UCI repository at `https://archive.ics.uci.edu/ml/datasets/air+quality`. The covariates include information collected from sensors and environmental parameters, such as temperature and humidity, and we aim to predict the CO gas sensor measurement. The true_label is collected using a certified reference analyzer. While the given_label is collected through an Air Quality Chemical Multisensor Device, which is susceptible to sensor drift that can affect the sensors' concentration estimation capabilities.

**Metaphor Novelty dataset**: This dataset is derived from data provided by `http://hilt.cse.unt.edu/resources.html`. The regression task is to predict metaphor novelty scores given two syntactically related words. We have used FastText word embeddings to calculate vectors for both words available in the dataset. The true_label is collected using expert annotators, and the given_label is the average of all five annotations collected through Amazon Mechanical Turk.

**Stanford Politeness Dataset (Stack edition)**: This dataset is derived from data provided by `https://convokit.cornell.edu/documentation/stack_politeness.html`. The regression task is to predict the level of politeness conveyed by some text, in this case requests from the Stack Exchange website. The given_label is randomly selected from one of five human annotators that rated the politeness of each example, while the median of all five annotators' politeness ratings is considered as the true_label. As covariates for our regression models, we use numerical covariates obtained by embedding each text example

via a pretrained Transformer network from the Sentence Transformers package Reimers and Gurevych (2019).

**Stanford Politeness Dataset (Wikipedia edition)**: This dataset is derived from data provided by `https://convokit.cornell.edu/documentation/wiki_politeness.html`. The regression task, feature embeddings, given_label, and true_label are the same as those in the Stanford Politeness Dataset (Stack edition), but here the text is a collection of requests from Wikipedia Talk pages.

**qPCR Telomere**: This dataset is a subset of the dataset generated through an R script provided by `https://zenodo.org/record/2615735#.ZBpLES-B30p`. It is a simple regression task where independent covariates are taken from a normal distribution, and the true_label is generated by $f(x_i)$. The given_label is defined as true_label + error. While this is technically a simulated dataset, the simulation was specifically aimed to closely mimic data noise encountered in actual qPCR experiments.

## Appendix C. Additional Benchmark Comparisons

For more comprehensive evaluation, we additionally compare against a number of other model-agnostic baseline approaches to detect errors in numeric data. Each baseline here produces a veracity score which can be used to rank data by their likelihood of error, as done for our proposed methodology.

We evaluated these alternative scores following the same procedure (same metrics and datasets) from our real dataset benchmark. No data filtering procedure was applied for any of these methods, models were simply fit via $K$-fold cross-validation to produce out-of-sample predictions for the entire dataset, which were then used to compute veracity scores under each approach. Table 8 below shows that none of these alternative methods are consistently superior to the straightforward residual veracity score studied in our other evaluations.

Here are descriptions of the baseline methods we considered as alternative veracity scores:

**Relative Residual.** This baseline veracity score is defined as:

$$\exp\left(-\frac{|y - \hat{y}|}{|y| + \epsilon}\right) \tag{2}$$

where $\epsilon = 1e - 6$ is a small constant for numeric stability. The *relative residual* rescales the basic residual by the magnitude of the target variable $Y$, since values of $Y$ with greater magnitude are often expected to have larger residuals.

**Marginal Density.** This baseline veracity score is defined as: $\hat{p}(y)$, the (estimated) density of the observed $y$ value under the marginal distribution over $Y$. Here we use kernel density estimates, and this approach does not consider the feature values $X$ at all. The *marginal density* score is thus just effective to detect $Y$ values that are atypical in the overall dataset (i.e. overall outliers rather than contextual outliers).

**Local Outlier Factor.** This baseline veracity score attempts to better capture datapoints which have either high residual or low marginal density, since either case may be indicative

of an erroneous value. First we form a 2D scatter plot representation of the data in which one axis is the residual: $|\hat{y}-y|$ and the other axis is the original $y$-value. Intuitively, outliers in this 2D space correspond to the datapoints with abnormal residual or $y$-value. Thus we employ the *local outlier factor* (LOF) score to quantify outliers in this 2D space, employing this as an alternative veracity score (Breunig et al., 2000).

**Outlying Residual Response (OUTRE).**   This baseline veracity score is similar to the Local Outlier Factor approach above, and identifies outliers in the same 2D space in which each datapoint is represented in terms of its residual and $y$-value. Instead of the LOF score, here we score outliers via their average distance to the k-nearest neighbors of each datapoint Kuan and Mueller (2022b), and use the inverse of these distances as an alternative veracity score.

**Discretized.**   This baseline veracity score is defined by reformulating the regression task as a classification setting, and then applying methods that are effective to detect label errors in classification. More specifically, we discretize the $y$ values in the dataset into 10 bins (defined by partioning the overall range of the target variable). For each bin $k$, we construct a model-predicted "class" probability for that bin proportionally to: $exp(-|\hat{y}-c_k|)$ where $c_k$ is the center of bin $k$. After renormalizing these probabilities to sum to 1 over $k$, this offers a straightforward conversion of regression model outputs to predicted class probabilities if the bins are treated as the possible values in a classification task. Finally, we apply Confident Learning with the self-confidence veracity score to produce a veracity score for each datapoint Northcutt et al. (2021); Kuan and Mueller (2022a). This approach uses the given class label (identity of the bin containing each $y_i$) and model-predicted class probabilities to identify which datapoints are most likely mislabeled.

Table 8: Evaluation of alternate scoring methods across various metrics. The results are reported as the average across all datasets and across all models, as discussed in Section B.

| scoring method | AUPRC | AUROC | lift_at_100 | lift_at_num_errors |
|---|---|---|---|---|
| residual | 0.42 | 0.71 | 5.78 | 4.96 |
| relative residual | 0.36 | 0.66 | 4.61 | 4.32 |
| marginal density | 0.19 | 0.57 | 0.58 | 1.47 |
| local outlier factor | 0.20 | 0.64 | 1.99 | 2.03 |
| OUTRE | 0.42 | 0.73 | 5.76 | 4.97 |
| discretised | 0.31 | 0.66 | 4.85 | 3.01 |

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
