# OpenReview forum: "Detecting Errors in a Numerical Response via any Regression Model"
_DMLR — Accepted by DMLR_

### Review · Reviewer_VGUp · 2023-12-23

**Recommendation:** 3
**Confidence:** 1

**Summary Of Contributions:**

This paper presents a model-agnostic approach for identifying incorrect values in numerical datasets. This method, utilizing any fitted regressor, distinguishes between anomalies and normal data variations by considering various uncertainties. The approach comes with theoretical guarantees and outperforms existing methods like conformal inference in error detection. The effectiveness of this method is demonstrated through a new benchmark involving five regression datasets with real-world numerical errors and additional simulation studies, showing superior precision and recall in identifying incorrect values.

**Audience:**

Yes

**Broader Impact Concerns:**

I have no concerns

**Claims And Evidence:**

yes

**Datasets And Benchmarks:**

yes

**Extended Submissions:**

not applicable

**Limitations:**

[-] When the error or noise data are detected, how it will benefit downstream applications, such as classification or prediction? Some experiments to empirically support this aspect will be appreciated.

[-] A clear problem statement would further improve this paper. For example, what is the scope of application of the proposed method? For example, 2-d image data, video data, natural language data, time series data. While authors mention the labels are continuous, more explanation would be appreciated.

[-] What is the relationship between anomaly detection and error value detection? It seems anomaly detection also aims to detect abnormal / corrupted values, e.g., in industrial time series data. Since it is also a growing research area, more discussion about this will be appreciated.

[-] The limitations and future work are missing from the manuscript. The authors might need to discuss them to improve the readability and benefit follow-up studies.

[-] For real datasets, it is unclear how to obtain groundtruth/accurate data to evaluate the proposed method. Because basically all real data contain certain levels of noises or errors. Or, the authors assume the data is clean, and constructed a corrupted dataset by adding some noises? More explanation would be appreciated.

[-] More related work should be discussed.

[-] Another suggestion is that adding a figure to present the overall framework would improve the readability of this paper. For example, the fig could contain an example of data record, the proposed error detection method, the output, and probably a few possible applications based on the output data.

[-] The writing can be improved. For example, in the third paragraph in Introduction, it seems “veracity score” is a metric proposed by this paper. However, “score” is proposed in the fifth paragraph, which contradicts with prior content.

[-] The resolution of Fig.1 can be improved.

--- update after author responses ----
Most of my concerns are resolved now, therefore, I have raised the score.

**Requested Changes:**

Please see the weaknesses below.

**Strengths And Weaknesses:**

[+] The proposed method has no assumption about the model category and therefore can be adapted to many application scenarios.

[+] Both theoretical results and empirical analysis are provided.

[+] The benchmark designed by this paper could benefit future studies in this area.

---

### Review · Reviewer_uCLR · 2024-01-03

**Recommendation:** 3
**Confidence:** 3

**Summary Of Contributions:**

In this paper the authors tackle the problem of error detection in numerical data. In particular, they propose two veracity scores that are model-agnostic and can be used to identify errors in numerical datasets that consist of covariates and labels. The veracity scores account for epistemic and aleatoric uncertainty by normalizing residuals with the corresponding uncertainty scores. To avoid noise in heavily corrupted data, the authors propose a pruning step where most noisy datapoints are iteratively removed. The authors show that under certain statistical conditions the proposed approach outperforms simple residual-based scores. On five real-world datasets the proposed method is shown to be more effective than the baseline.

**Audience:**

Yes

**Claims And Evidence:**

The claims in the paper are accurate once the problem is defined properly. The title and the introduction overclaim the scope and should be narrowed down.

**Datasets And Benchmarks:**

Datasets are made available.

**Extended Submissions:**

NA

**Limitations:**

W1- The paper considers a narrow area of data errors. The title of the paper and the introduction first talk about very general error detection while the proposed approach considers a narrow type of numerical data errors.

W2- The paper is not clear about where the errors are supposed to reside. Whether every column will become Y or whether there are dedicated Y and X columns. Also in the description of the experiments this is not clear.

W3 En par with 1, the authors also overlook a large body of work on error detection
https://www.vldb.org/pvldb/vol9/p993-abedjan.pdf
https://chu-data-lab.cc.gatech.edu/data-cleaning-for-ml/
https://raulcastrofernandez.com/papers/raha.pdf
https://vldb.org/pvldb/vol5/p1674_tamraparnidasu_vldb2012.pdf
to name a few

**Requested Changes:**

- Narrow down the scope and position with regard to related work in data cleaning as mentioned under limitations.

**Strengths And Weaknesses:**

S1- An interesting formalization of errors and epistemic and aleatoric outliers. The proposed scores are model-independent.
S2- The authors conduct experiments on real-world datasets and provide the corresponding benchmark.

---

### Review · Reviewer_iZZC · 2024-01-17

**Recommendation:** 3
**Confidence:** 1

**Summary Of Contributions:**

This work introduced new veracity scores that incorporate both epistemic and aleatoric uncertainties, enabling the error detection procedure to more effectively distinguish between genuine anomalies and natural data fluctuations. This approach enhances the reliability of identifying errors in numerical data.

**Audience:**

Yes

**Claims And Evidence:**

Yes

**Datasets And Benchmarks:**

N/A

**Extended Submissions:**

N/A

**Limitations:**

- The performance of error detection is significantly influenced by the choice of the regression model, which may not always align perfectly with the dataset characteristics.
- The method's effectiveness and computational efficiency in high-dimensional data scenarios remain unclear.
- The presentation of the work could be improved.

**Requested Changes:**

- Further discussion on the computational complexity of the proposed metric w.r.t. data dimensions: As the dimensionality of the dataset increases, the complexity of the model and the computational resources required also increase. The proposed approach in high-dimensional spaces and its scalability need to be explored further.
- Intuition of the proposed metric: It would also be nice to include a figure to provide the intuitions behind the proposed metric.

**Strengths And Weaknesses:**

- novel veracity scores
- theoretical guarantees and improved precision/recall
- broad applicability and model agnostic